# Microbial Interactions as Drivers of a Nitrification Process in a Chemostat

**DOI:** 10.3390/bioengineering8030031

**Published:** 2021-02-25

**Authors:** Pablo Ugalde-Salas, Héctor Ramírez C., Jérôme Harmand, Elie Desmond-Le Quéméner

**Affiliations:** 1LBE, INRAE, Université de Montpellier, 11100 Narbonne, France; jerome.harmand@inrae.fr (J.H.); elie.le-quemener@inrae.fr (E.D.-L.Q.); 2Departamento de Ingeniería Matemática, Centro de Modelamiento Matemático (CNRS UMI 2807), Universidad de Chile, Santiago, Chile; hramirez@dim.uchile.cl

**Keywords:** microbal interactions, microbial growth rate, bifurcation analysis, generalized Lotka–Volterra, chemostat theory, optimal control

## Abstract

This article deals with the inclusion of microbial ecology measurements such as abundances of operational taxonomic units in bioprocess modelling. The first part presents the mathematical analysis of a model that may be framed within the class of Lotka–Volterra models fitted to experimental data in a chemostat setting where a nitrification process was operated for over 500 days. The limitations and the insights of such an approach are discussed. In the second part, the use of an optimal tracking technique (developed within the framework of control theory) for the integration of data from genetic sequencing in chemostat models is presented. The optimal tracking revisits the data used in the aforementioned chemostat setting. The resulting model is an explanatory model, not a predictive one, it is able to reconstruct the different forms of nitrogen in the reactor by using the abundances of the operational taxonomic units, providing some insights into the growth rate of microbes in a complex community.

## 1. Introduction

Microbial communities and their interactions play a central role in the understanding of microbial ecosystems [1], and a current challenge is integrating genetic sequencing data in a deterministic modelling framework [2,3]. Using the terminology from the thorough review in current methodologies on the deterministic modelling approaches of microbial community dynamics presented by Song et al. [4], this articles deals with population-based approaches where species are taken as the interacting units.

The classical ecological concept of species and niche in the microbial world is an elusive one: in the macro world one can clearly differentiate one species from another for reproductive reasons and their ability to give birth to offspring. In the case of bacteria and archea, reproduction goes simply by binary fission and exchange of some functional genes (e.g., the ability to synthesize or metabolize substances) can be acquired in evolutionary scale through lateral gene transfer [5]. Therefore as an ecological problem is hard to define precisely the ‘niche’ of ‘microbial species’. These obstacles can be circumvented by considering the microbiologist concept of operational taxonomic unit (OTU) based on the clustering of organisms sharing similar sequences of the 16S rDNA marker gene. In the past years considerable efforts have been made to measure the bacterial community composition. Tests such as fluorescence in situ hybridization (FISH), polymerase chain reaction (PCR) dependent techniques, and PCR independent techniques for the analysis of DNA have become a standard tool for studying microbial diversity [6]. The contribution of this article is a new method to integrate the microbial community measurements in chemostat models, based on any sequencing or fingerprinting technique that can quantify the species abundances over time. In other words, while most models used in bioengineering are functional—in the sense they consider only one species per biological reaction considered—this work is an attempt to merge classical population-based models used in ecology and those used by engineers in biotechnology.

Interactions lie at the heart of ecology. Lotka [7] and Volterra [8], independently, presented a 2 dimensional dynamical system to model prey-predator relationships, now known as the Lotka–Volterra (LV) equations. The model is very rich from a mathematical standpoint, and is also a classic equation to study in Mathematical Ecology [9]. Extensions of the Lotka–Volterra model have derived what is now known as generalized Lotka–Volterra (gLV) models [10] shown in Equation (1):(1)$$\begin{array}{c} \dot{x_{i}}=\mu_{i}\left(1+\sum_{j=1}^{n}a_{ij}x_{j}\right)x_{i}\;i\in \left\{1,\ldots ,n\right\} \end{array}$$
where $x_{i}$ represents the species abundance, $\mu_{i}$ the intrinsic growth rate of the species, and the terms $a_{ij}$ reflect the effect of OTU *j* on the growth of OTU *i*. The equation states that the growth rate of $x_{i}$ is proportional to $x_{i}$, but this proportionality constant depends on its intrinsic growth rate multiplied by the sum of all interactions affecting it. Note that if there are no type of interactions ($a_{ij}=0$), one recovers *n* uncoupled linear differential equations, and thus the solution becomes $x_{i}\left(t\right)=x_{i}\left(0\right)\exp\left(\mu_{i}t\right)$, that is exponential growth on time.

The diagonal terms $a_{ii}$ are known as intraspecies interaction, while the off diagonal terms are known as the pairwise interspecies interactions. Noting the signs of pairs $\left(a_{ij},a_{ji}\right)$, the classical ecological relationships of mutualism or cooperation (+,+), commensalism (+,0), predation or parasitism (+,−), competition (−,−), and ammensalism (−,0) can be recovered [11]. Model (1) has been thoroughly analysed, even when the coefficients $\mu_{i}$ and $a_{ij}$ are time dependent and exhibit periodicity (which models seasonal traits) [12,13]. The gLV model has been used in microbial ecology to some degree of success to study the gut microbiome of mice infected with *C. difficile* [14]. However, the quadratically growing number of parameters to describe interactions naturally entails problems of identifiability if the data set is not large enough, or the system has not been sufficiently perturbed. On a more conceptual ground, the interaction coefficients of a gLV model do not represent mechanistically anything, so even if a model correctly predicts the microbial community dynamics, it might not add to the understanding of what could be physically or biologically taking place. These observations led us to develop what can be considered the core contribution of this article, which is to study the growth rate of each species in a mixed culture: we reconstruct the shape of their growth rate, instead of trying to fit a particular function (such as the gLV equations). As Monod himself commented when developing the growth law that bears their name is that any function with the same shape (monotone, concave, and bounded on the substrate) would have served [15], in this spirit we formulate the question: what is the shape of the growth functions of multiple species developing together?

As a departing point, the work of Dumont et al. [16] is presented in Section 2. They modelled a chemostat experiment where nitrification takes place by considering a glV model coupled with a substrate limited growth expression ($\mu_{i}$ is no longer a constant, but the classic Monod expression) and fitted their model using absolute abundances of the major OTU identified by molecular fingerprints obtained by single-strand conformation polymorphism (SSCP). Section 3 inspects the model through a mathematical analysis. Some interesting outputs of this analysis are that the number of possible equilibrium points grows exponentially with the number of species, coexistence can be achieved within the same functional group, and bi-stability may arise. In Section 4, the concept of interaction function is developed such that it generalizes the gLV model. For approximating the interaction function a method of optimal control theory was adapted: The growth rate of each species is modulated by a constrained regular control of the system, thus the growth rate of each OTU is corrected in order to fit the experimental data. The regular control is composed of a feedback part on the species state variable, and a feed forward part, or tracking, on the measurement of abundances of each species; the method involves solving state-dependent Ricatti equations [17]. In Section 5, the methodology from Section 4 is applied to the data from the experiments performed by Dumont et al. [18] and not just to the most abundant species as it was the case of the model analysed in Section 3 [16]. This approach explicitly assumes that dynamics of complex ecosystems are driven by interactions, that are the results of feedback loops of each species on the growth rate of others. The method shows that by following the community dynamics one can propose a growth rate that reconstructs the substrates dynamics, however this cannot be considered a predictive model, but rather a explicatory model. The article ends with a discussion on the scope of applicability and perspectives of the method.

## 2. Model Definition

Notations used throughout the article:
*n*: the number of OTU considered.$n_{i},\;i\in \left\{1,2\right\}$: the number of OTU in functional group $G_{i}$. In the example $G_{1}$ corresponds to ammonia oxidizing bacteria (AOB) and $G_{2}$ corresponds to nitrite oxidizing bacteria (NOB).Let *m* be an interger then [m]:={1,…,n}.$x_{i}$: is the concentration of OTU *i* measured in [g/l]. i∈[n].*x*: vector ${\left(x_{1},\ldots ,x_{n}\right)}^{⊤}$.$s_{1}$: concentration of substrate 1 in [g/l]. In the example $s_{1}$ represents ammonium.$s_{2}$: concentration of substrate 2 in [g/l]. In the example $s_{2}$ represents nitrite.$s_{3}$: concentration of substrate 3 in [g/l]. In the example $s_{3}$ represents nitrate.$s_{in}$: entry concentration of substrate 1 in [g/l]. May depend on time $s_{in}=s_{in}\left(t\right)$.*s*: vector ${\left(s_{1},s_{2},s_{3}\right)}^{⊤}$. Referred to as metabolites.$I_{i}\left(t,x\right)$: Interaction function of OTU i∈{1,…,n}.$\mu_{i}\left(s,x\right)$: growth function of OTU i∈{1,…,n}.$\mu =(\mu_{1}\left(s,x\right),\ldots ,\mu_{n}\left(s,x\right))$ vector containing the growth function of every OTU.*D*: dilution rate of the continuous reactor in [1/day]. May depend on time D=D(t).$y_{i}$: yield of grams of OTU *i* formed per gram of substrate consumed.$y_{s_{i}/x_{j}}$: yield of grams of substrate $s_{i}$ consumed/produced per gram of OTU *j* formed. If negative it represents consumption, if positive it represents production.*Y*: matrix containing all yields such that $Y_{ij}=y_{s_{i}/x_{j}}$.For integers $m_{1}$ and $m_{2}$ and a∈R, $a_{m_{1}\times m_{2}}$ represents a matrix of $m_{1}$ rows and $m_{2}$ columns with *a* in every entry.Let *m* be an integer then $I_{m}$ is the identity matrix of size *m*.Let *M* be a matrix, then $M_{i•}$ represents the *i*-th row of matrix *M*.Let *S* be a finite set with m∈N elements. Then |S|:=m.Given a vector $v=\left(v_{1},\ldots ,v_{n}\right)\in R^{n}$, the function diag(v) stands for:
(2)$$\begin{array}{c} \begin{array}{cc} \mathrm{diag}:R^{n}\to & M_{n\times n}\left(R\right)\\ \; & \\ v\to & \left(\begin{array}{cccc} v_{1} & 0 & \ldots & 0\\ 0 & v_{2} & \ddots & \vdots \\ \vdots & \ddots & \ddots & 0\\ 0 & \ldots & 0 & v_{n} \end{array}\right) \end{array} \end{array}$$

### 2.1. Stoichiometric Equations

A cascade (bio)reaction process is considered. Suppose *n* different OTU are present in the chemostat. A two step cascade reaction refers to the situation where a group of microorganisms ($G_{1}\subset \left[n\right]$) consumes a substrate $s_{1}$ and produces $s_{2}$ and biomass, while another group of microorganisms ($G_{2}\subset \left[n\right]$) consumes $s_{2}$ and produces $s_{3}$ and biomass. $G_{1}$ and $G_{2}$ are called functional groups. The number of organisms in each functional will be denoted $n_{1}$ and $n_{2}$, respectively, that is $|G_{1}|=n_{1}$ and $|G_{2}|=n_{2}$. This work treats the case when $G_{1}$ and $G_{2}$ are disjoint sets:

**Hypothesis** **1** **(H1).**
*Sets $G_{1}$ and $G_{2}$ satisfy: $G_{1}\cap G_{2}=\emptyset$ and $G_{1}\cup G_{2}=\left[n\right]$.*


The situation is described as simplified Reactions (R1) and (R2). The reactions are simplified in the sense that they do not attempt to represent a balanced chemical reaction, rather they represent the direction of the bioprocess and the proportions of different consumed and formed compounds of interest. The terms $y_{i}$ are known as yields, they represent the quantity of g of biomass produced per g of substrate consumed by OTU *i*. For example, in the case of reaction (R1), one gram of $s_{1}$ is consumed, one gram of $s_{2}$ and $y_{x_{i}/s_{1}}$ grams of dry biomass of OTU *i* are produced.
(R2)$$\begin{array}{c} s_{1}\overset{\mu_{i}\left(s,x\right)}{⟶}s_{2}+y_{i}x_{i}\;\forall i\in G_{1} \end{array}$$
(R2)$$\begin{array}{c} s_{2}\overset{\mu_{i}\left(s,x\right)}{⟶}s_{3}+y_{i}x_{i}\;\forall i\in G_{2} \end{array}$$

However for expressing the system of differential equations further below, the terms $y_{s_{i}/x_{j}}$ are used. They express the grams of substrate $s_{i}$ consumed (negative sign) or produced (positive sign) per gram of OTU *j* formed. They are related to $y_{i}$ as seen in Table 1. This defines the stoichiometry matrix $Y\in R^{3\times n}$, such that $Y_{ij}=y_{s_{i}/x_{j}}$.

Furthermore, for each i∈[n], OTU *i* is characterized by its process rate (also known as growth function) $\mu_{i}\left(s,x\right)$. Notice that for being as generic as possible, the growth rate may be a function of the whole state in order to model the influence of all OTU on the growth rates of others.

An example of this process is the nitrification process where group $G_{1}$ is known as Ammonia oxidizing Bacteria (AOB), and group $G_{2}$ is known as Nitrite oxidizing Bacteria (NOB) [19].

### 2.2. Mass Balance Equations

Consider the scenario of a continuous and homogeneous reactor: the input flow is the same as the output flow, with a dilution rate *D*. The input flow contains a concentration $s_{in}$ of substrate $s_{1}$. Each OTU grows at a rate $\mu_{i}\left(s,x\right)$. System (3) represents this situation. A specific case of $\mu_{i}\left(s,x\right)$ is given in the next subsection.
(3)$$\begin{array}{c} \begin{array}{cc} \dot{x_{i}}= & \left(\mu_{i}\left(s,x\right)-D\right)x_{i}\;\forall i\in \left[n\right]\\ \dot{s_{1}}= & \left(s_{in}-s_{1}\right)D-\sum_{i\in G_{1}}\frac{1}{y_{i}}\mu_{i}\left(s,x\right)x_{i}\\ \dot{s_{2}}= & -s_{2}D+\sum_{i\in G_{1}}\frac{1}{y_{i}}\mu_{i}\left(s,x\right)x_{i}-\sum_{i\in G_{2}}\frac{1}{y_{i}}\mu_{i}\left(s,x\right)x_{i}\\ \dot{s_{3}}= & -s_{3}D+\sum_{i\in G_{2}}\frac{1}{y_{i}}\mu_{i}\left(s,x\right)x_{i} \end{array} \end{array}$$

System (3) can also be written in a more compact form using the stoichiometric matrix *Y* and the diag operator.
(4)$$\begin{array}{cc} \dot{x}= & \mathrm{diag}(\mu \left(x,s\right)-D_{n\times 1})x \end{array}$$
(5)$$\begin{array}{cc} \dot{s}= & \left({\left[\begin{array}{ccc} s_{in} & 0 & 0 \end{array}\right]}^{⊤}-s\right)D+Y\mathrm{diag}\left(\mu \left(x,s\right)\right)x \end{array}$$

### 2.3. Kinetic Equations

In the work of Dumont et al. [16], the growth rates seen in Equation (6) were calibrated against experimental data for the two most abundant OTU of each functional group.
(6)$$\begin{array}{c} \begin{array}{cc} \mu_{i}\left(s,x\right) & ={\bar{\mu }}_{i}\frac{s_{1}}{K_{i}+s_{1}}\left(1+\sum_{j\in [n]}a_{ij}x_{j}\right)\;\forall i\in G_{1}\\ \mu_{i}\left(s,x\right) & ={\bar{\mu }}_{i}\frac{s_{2}}{K_{i}+s_{2}}\left(1+\sum_{j\in [n]}a_{ij}x_{j}\right)\;\forall i\in G_{2} \end{array} \end{array}$$

The term $\left(1+\sum_{j\in [n]}a_{ij}x_{j}\right)$ accounts for pairwise interactions affecting the growth rate of each OTU, while the term ${\bar{\mu }}_{i}\frac{s_{j}}{K_{i}+s_{j}}$ is a Monod growth expression, where ${\bar{\mu }}_{i}$ represents the maximum growth rate, and $K_{i}$ the half saturation constant [15]. Note that if every $a_{ij}=0$, then one recovers a classic substrate limited growth. Let *A* denote the matrix with entries $a_{ij}$ hereafter referred to as the interaction matrix. Dumont et al. did not analyse their model but simply provided several simulations using parameter values identified from experimental data. The following section of this article deals with the mathematical analysis of model (3) with growth rates given by (6).

## 3. Mathematical Analysis

The system of Equation (3) is defined in the region
$$\begin{array}{c} \Omega :=\{\left(x_{1},\ldots ,x_{n},s_{1},s_{2},s_{3}\right)\in R^{n+3}|x_{1},\ldots ,x_{n},s_{1},s_{2},s_{3}\geq 0\} \end{array}$$

First, sufficient conditions on the interaction matrix for the system to be well posed are established: meaning that solutions remain bounded and non-negative in time, this ultimately implies that the solution exists for every t≥0 [20].

Second, the equilibria of the system are derived. Possible equilibrium points for this system grow exponentially with the number of OTU considered (*n*). Stability is not analytically addressed, a numerical scheme calculating every equilibrium point and the system’s Jacobian eigenvalues at the equilibrium point was implemented for studying the system.

### 3.1. Properties of the System

A bound on the norm of the interaction matrix that depends on the initial conditions and parameters one establishes that solutions will remain positive and bounded.

**Lemma** **1.**
*For initial conditions $(x_{1}\left(0\right),\ldots ,x_{n}\left(0\right),s_{1}\left(0\right),s_{2}\left(0\right),s_{3}\left(0\right)))\in \Omega$, there exists positive scalars $M_{1}$, $M_{2}$, and $M_{3}$ such that solutions to (3) satisfy the following inequalities:*
(7)$$\begin{array}{cc} \; & \sum_{i\in G_{1}}\frac{1}{y_{i}}x_{i}+s_{1}\leq M_{1} \end{array}$$
(8)$$\begin{array}{cc} \; & \sum_{i\in G_{2}}\frac{1}{y_{i}}x_{i}+s_{1}+s_{2}\leq M_{2} \end{array}$$
(9)$$\begin{array}{cc} \; & s_{1}+s_{2}+s_{2}\leq M_{3} \end{array}$$


The proof can be seen in Appendix A. A bound on the norm of *A* is found such that every matrix *A* respecting the bound, guaranties that Ω is a positively invariant set.

**Lemma** **2.***For initial conditions $(x_{1}\left(0\right),\ldots ,x_{n}\left(0\right),s_{1}\left(0\right),s_{2}\left(0\right),s_{3}\left(0\right)))\in \Omega$, there exists a constant M>0 such that for every matrix A satisfying ${∥A∥}_{\infty }\leq M$, the solutions of system (3) with growth rates given by (6), remain in* Ω *and are bounded*.


The proof can be seen in Appendix A.

The importance of Lemma 2 is that by restricting the norm of matrix *A* the system is well-posed, meaning that the solutions can have biological and physical sense (there is no such thing as negative concentrations). Particularly, one has that ${∥A∥}_{\infty }=\max_{1\leq i\leq m}\sum_{j=1}^{n}\left|a_{ij}\right|$, which in this context implies that a bound on the sum of the absolute value of the interaction terms that affects each species allows the system to be well-posed. Note, however, this is a sufficient condition, thus the range of values matrix *A* can sustain for the system to remain well-posed may be considerably larger.

### 3.2. Equilibrium Points

In this section, analytical expressions for equilibrium points are shown. However, no analytic expression concerning the stability of such points is presented. In the following pages the reader will appreciate that the expressions of the equilibrium points are not simple, consequently replacing them in a 5×5 block matrix and calculating eigenvalues resisted an algebraic treatment. To answer the question of stability a numeric scheme is used by evaluating the Jacobian at the equilibrium point. At the end of the section an algorithm is provided for exploring all the possible equilibria. All the computations for deriving the equations of this section can be found in Appendix B.

Let f(s) be such that,
(10)$$\begin{array}{c} f_{i}\left(s\right)=\left\{\begin{array}{c} {\bar{\mu }}_{i}\frac{s_{1}}{K_{i}+s_{1}}\;\forall i\in G_{1}\\ {\bar{\mu }}_{i}\frac{s_{2}}{K_{i}+s_{2}}\;\forall i\in G_{2} \end{array}\right. \end{array}$$

Then $\mu \left(x,s\right)=\mathrm{diag}\left(f\left(s\right)\right)\left(1_{n\times 1}+Ax\right)$

Thus, system (3) is rewritten as follows.
(11)$$\begin{array}{cc} \dot{x}= & \mathrm{diag}(\mu \left(x,s\right)-D_{n\times 1})x \end{array}$$
(12)$$\begin{array}{cc} \dot{s_{1}}= & \left(s_{in}-s_{1}\right)D+Y_{1•}\mathrm{diag}\left(\mu \left(x,s\right)\right)x \end{array}$$
(13)$$\begin{array}{cc} \dot{s_{2}}= & -s_{2}D+Y_{2•}\mathrm{diag}\left(\mu \left(x,s\right)\right)x \end{array}$$
(14)$$\begin{array}{cc} \dot{s_{3}}= & -s_{3}D+Y_{3•}\mathrm{diag}\left(\mu \left(x,s\right)\right)x \end{array}$$

**Definition** **1.**
*An equilibrium point (or steady state) is a point $(x^{eq},s^{eq})\in \Omega$ so that the right hand side of Equations (11)–(14) equals zero.*


Observe that equilibrium points are by definition non-negative so the state variables can have physical meaning. For studying the cases where $x^{eq}$ contains zero valued entries, the set of non-active coordinates is defined as follows:

**Definition** **2.**
*Given an equilibrium point $\left(x^{eq},s^{eq}\right)$ of system (3), then the set of non-active coordinates J⊂{1,…,n} is defined as: $J=\{j_{1},\ldots ,j_{m}\;:\;x_{j_{i}}^{eq}=0,\;i\in \left[m\right]\}$. $n_{1}^{act}$ and $n_{2}^{act}$ denote the number of positive entries of $x^{eq}$ of functional groups $G_{1}$ and $G_{2}$, respectively. $n^{act}=n-m$ denotes the total number of positive entries of $x^{eq}$. The active point $x^{act}\in R^{n^{act}}$ is defined by the positive entries of $x^{eq}$. Analogously, the functions $f^{act}\left(s\right)$ and $\mu^{act}\left(x,s\right)$ are defined by the positive entries of $x^{eq}$. The active interactions $A^{act}$ is defined as the matrix A without the J rows and columns. The active stoichiometry matrix $Y^{act}$ is the matrix Y without the J columns.*


In order to derive the equilibrium points, it is desirable an invertible $A^{act}$ matrix. Therefore in what follows of the work it is assumed that matrix *A* and some of its submatrices have an inverse, this is stated properly in Hypothesis 2.

**Hypothesis** **2** **(H2).**
*Let A be the interaction matrix of size n∈N and S be a proper subset of [n] with |S|=m. Then the matrix $B\in R^{\left(n-m)\times (n-m\right)}$ defined by taking out the S rows and columns of matrix A is invertible.*


Assuming Hypothesis 2 a formula for the active points is derived from Equation (11):(15)$$\begin{array}{c} x^{act}={\left(A^{act}\right)}^{-1}\left(\mathrm{diag}{\left(f^{act}\left(s\right)\right)}^{-1}D_{n^{act}\times 1}-1_{n^{act}\times 1}\right) \end{array}$$

Note as well that at the equilibrium, $s_{3}$ can be defined in terms of $s_{1}$, $s_{2}$ and $s_{in}$. This is done by adding Equations (12)–(14) which gives:(16)$$\begin{array}{c} s_{in}=s_{1}+s_{2}+s_{3} \end{array}$$

#### 3.2.1. Both Functional Groups Are Present

The case where in each functional group remains at least one OTU is represented by Hypothesis 3.

**Hypothesis** **3** **(H3).**
*The set J satisfies $G_{1}⊄J,\;G_{2}⊄J$.*


By replacing Equation (15) in Equation (12) $s_{2}$ can be written as a function of $s_{1}$:(17)$$\begin{array}{c} s_{2}=\frac{s_{1}}{b_{1}s_{1}^{2}+b_{2}s_{1}+b_{3}} \end{array}$$

Then by replacing (17) in Equation (13), one gets a fourth degree polynomial for $s_{1}$.
(18)$$\begin{array}{c} a_{4}s_{1}^{4}+a_{3}s_{1}^{3}+a_{2}s_{1}^{2}+a_{1}s_{1}+a_{0}=0 \end{array}$$

Formulae for coefficients $b_{1},b_{2},b_{3},a_{0},a_{1},a_{2},a_{3},a_{4}$ can be found in Appendix B.

The equilibrium point can be calculated from the solutions of the system of Equations (15)–(18) with non negative coordinates. If the system only provides solutions with at least one negative entry then the set J cannot define an equilibrium point.

#### 3.2.2. Washout of $G_{2}$

The washout of $G_{2}$ is equivalent to Hypothesis 4.

**Hypothesis** **4** **(H4).**
*$G_{2}\subset J$ and $G_{1}⊄J$.*


Under this case note that $f^{act}\left(s\right)$ depends only on $s_{1}$. Therefore when Equation (15) is replaced in (12), one obtains a quadratic equation for $s_{1}$:(19)$$\begin{array}{c} a_{2}^{\prime }s_{1}^{2}+a_{1}^{\prime }s_{1}+a_{0}^{\prime }=0 \end{array}$$
where $a_{i}^{\prime }$ can be found in Appendix B.

Since $x_{i}=0\;\forall i\in G_{2}$ then from Equation (14).
(20)$$\begin{array}{c} \dot{s_{3}}=0=-s_{3}D \end{array}$$
(21)$$\begin{array}{c} \Rightarrow s_{3}=0 \end{array}$$

In this case the equilibrium point can be calculated from the solutions of the system of Equations (15), (16), (A44) and (21) with non-negative coordinates. If the system only provides solutions with at least one negative entry then the set J cannot define an equilibrium point.

#### 3.2.3. Washout of $G_{2}$

The washout equilibria means $x_{i}=0$ for every $i\in \{1,\ldots ,n_{1}\}$. This is equivalent to Hypothesis 5. Note that the structure of a cascade reaction implies that if $G_{1}$ gets washed out, then so is $G_{2}$.

**Hypothesis** **5** **(H5).**
*$J=G_{1}\cup G_{2}$.*


From Equation (12), one gets
$$\begin{array}{c} s_{in}=s_{1} \end{array}$$
then (16) implies
$$\begin{array}{cc} s_{2} & =s_{3}=0 \end{array}$$

The equilibrium is then given by ${\left(\begin{array}{cccc} 0_{1\times n} & s_{in} & 0 & 0 \end{array}\right)}^{⊤}$.

All the former discussion leads to a potential number of $\left(2^{n_{1}}-1\right)\cdot \left(4\cdot \left(2^{n_{2}}-1\right)+2\right)+1$ different equilibria. Indeed:(22)$$\begin{array}{c} \overset{\begin{array}{c} \mathrm{nonempty}\;\mathrm{subsets}\\ \mathrm{of}\;G_{1} \end{array}}{\overbrace{\left(2^{n_{1}}-1\right)}}\left(\underset{\begin{array}{c} \mathrm{possible}\\ \mathrm{solutions}\;\mathrm{of}\\ \mathrm{Equation}\;\left(18\right) \end{array}}{\underbrace{4}}\cdot \overset{\begin{array}{c} \mathrm{nonempty}\;\mathrm{subsets}\\ \mathrm{of}\;G_{2} \end{array}}{\overbrace{\left(2^{n_{2}}-1\right)}}+\underset{\begin{array}{c} G_{2}\\ \mathrm{Washout} \end{array}}{\underbrace{2}}\right)+\overset{\mathrm{Washout}}{\overbrace{1}} \end{array}$$

### 3.3. Stability: Operating and Ecological Diagrams

In this subsection the stability of the equilibrium points is addressed. Operating and ecological diagrams are created from this stability analysis. Both are an illustrative way of representing the long term behaviour of a reactor depending on operating parameters, namely *D* and $s_{in}$: In a $D-s_{in}$ plane different zones representing the stability properties of system (3) are identified [21].

For checking local asymptotic stability of the equilibrium points, the Jacobian of the system is provided and evaluated at each of these points. The resulting matrix’s eigenvalues must have negative real part. A general formula for this Jacobian is presented in Appendix B (see (A70)).

Algorithm 1 summarizes this procedure.

**Algorithm 1:** Algorithm for Evaluating the Possible Equilibrium Points of System (3).

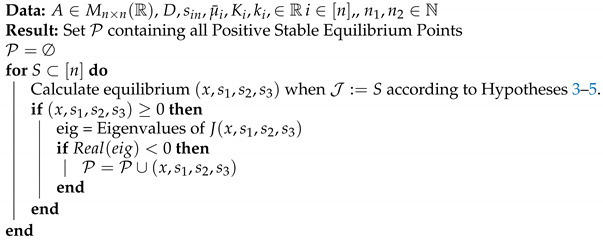



Operating and ecological diagrams are created by running Algorithm 1 for different pairs $\left(s_{in},D\right)$. In the case of operating diagrams [22] (OD) all the pairs $\left(s_{in},D\right)$ are regrouped such that the points of the set P represent when partial nitrification (PN), complete nitrification (CN), washout (WO), or a combination of them may arise [23]. PN refers to the state when nitrite ($s_{2}$) accumulates because the OTU of $G_{2}$ are washed out and thus no conversion from $s_{2}$ to $s_{3}$ takes place. On the contrary CN is when nitrate ($s_{3}$) accumulates because of the presence of OTU of $G_{2}$.

In the case of ecological diagrams (ED) the pairs are regrouped such that the points in the set P have the same non-active coordinates. In other words, instead of representing areas where either CN, PN or WO take place, ranges of pairs $\left(s_{in},D\right)$ where species coexist are represented. ED provide more information than OD, in the sense that one can deduce the latter from the former.

A first example using operating diagrams is presented to illustrate how adding interactions in a model consisting of 1 OTU in $G_{1}$ and 1 OTU in $G_{2}$ may lead to very different outcomes.

The important question of the existence of limit cycles was not resolved in this work. In the numerical analysis of this model at least one stable equilibrium was found for any choice of parameters. This obviously does not exclude the existence of limit cycles, but to what concerns the authors’ intuitions there always seems to be at least one stable equilibrium point.

#### 3.3.1. Case Study 1: 1 AOB and 1 NOB

Consider the case where $n_{1}=1$ and $n_{2}=1$. The operating diagrams when no interactions take place (A=0) and a non-zero interaction matrix are presented. When A=0 Algorithm 1 is no longer valid (A is not invertible), nevertheless the stability analysis is much simpler and is given in the Appendix B section. The interaction matrices are shown in Figure 1a,b, the rational behind the second choice was to force a very strong interaction of $x_{1}$ on $x_{2}$ and observe its effects. The biological reason behind a negative microbial interaction might be the release of a toxin by $x_{1}$ that affects $x_{2}$ [1], or in this case it might represent competition for oxygen; we stress the fact that gLV interactions do not explicitly account for mechanistically anything they just try to represent an ecological relationship taking place. The rest of parameters can be seen in Table 2.

The operating diagrams can be seen in Figure 2, note how partial nitrification (washout of $G_{2}$) of Figure 2b is much bigger when compared to Figure 2a. The shape of the PN region in Figure 2b is somewhat unintuitive, because at a constant dilution rate (0.24 day ${}^{-1}$ for example) and an increasing $s_{in}$, one passes from a PN zone, to a CN zone, and then back again to a PN zone. The mathematical explanation lies in the fact that $x_{1}$ also increases with $s_{in}$, and the affine part $\left(1+a_{21}x_{1}+a_{22}x_{2}\right)$ of the growth function of $x_{2}$ plays a bigger role than substrate limitation ($\frac{s_{2}}{s_{2}+K_{2}}$).

#### 3.3.2. Case Study 2: 2 AOB and 2 NOB

Case study 2 is based on Dumont et al. [16] model parameters. They proposed a distribution of parameters obtained from a Bayesian estimation method. Their fit describes well the dynamics of the two most abundant OTU in each functional group, but it still fails to capture the measured substrates dynamics. Kinetic parameters of case study 2 can be seen in Table 3. The estimated interaction matrix is shown in Figure 3a. A second matrix is presented, which is obtained by the sign change of coefficient $a_{11}$ (Figure 3b), and finally a third one is obtained by using a positive value for $a_{12}$ (Figure 3c). The idea is to show that qualitatively different outcomes can be obtained by changing one interaction at a time.

The ecological diagrams are presented in Figure 4, where the legend indicates the species surviving in the zone of the respective colour. In Figure 4b, the system exhibits bi-stability (it is represented in numbering as (1) and (2) of the different possible equilibriums). Note how every zone in Figure 4b has two stable equilibria, meaning that the outcome of the system is determined by its initial conditions, particularly interesting is the green zone where either $x_{1},x_{3}$ coexist or only $x_{2}$ remains, because in operational terms this means that either PN or CN may take place. When compared to Figure 4a, one can see that this change in the interactions of the microbial community can dramatically change the outcome of the reactor in a large operating zone.

One can see that coexistence in the same functional group is never attained in Figure 4a,b, whereas in Figure 4c $x_{1}$ and $x_{2}$, both AOB, coexist in either partial or complete nitrification. That means that the competitive exclusion principle [22] (CEP) does not hold. The CEP roughly states that if two species are growing on the same limiting resource, and their growth laws only depend non decreasingly on the resource, then only one of them will survive in the long run. This is interesting in light of reports on wastewater treatment plants where coexistence between species in nitrifying reactors has been shown [24], thus implying that a more complex growth law (as shown in here) or model structure involving other biological processes is required to include microbial diversity in mathematical models.

##### Remarks

Model (3) serves to illustrate that by considering a more complex growth rate that tries to model ecological interactions one might explain differences in reactors operating under similar conditions. It also shows a new mechanism by which the CEP no longer holds and which explains how multiple stable equilibria may appear. Since the gLV model discussed fails to completely capture the dynamics observed in the chemostat experiments [16], the next section proposes a new approach to study interactions.

## 4. Generalized Approach for Modelling Interactions

In the following an explanatory model (as opposed to a predictive model) is developed based on the hypothesis that interactions might be driving the nitrification process. In the previous sections interactions were modelled as an affine function of the OTU concentration that multiplies a substrate dependent growth equation. More generally the interaction function represents how the growth rate of species *i* is affected by the concentration of other species, *x*:

Given a vector ${\left(v_{1},\ldots ,v_{n}\right)}^{⊤}$ the interaction function I is denoted as:(23)$$\begin{array}{c} \begin{array}{cc} I:R_{+}^{n}\to & R_{+}^{n}\\ \; & \\ v\to & \left(\begin{array}{c} I_{1}\left(v\right)\\ I_{2}\left(v\right)\\ \vdots \\ I_{n}\left(v\right) \end{array}\right) \end{array} \end{array}$$

Let $f_{i}\left(s\right)$ be a bounded, positive, and continuous function of *s* (e.g., Monod, Haldane). The growth equation of OTU *i* becomes:(24)$$\begin{array}{c} \mu_{i}\left(s,x\right)=f_{i}\left(s\right)I_{i}\left(x\right) \end{array}$$

Note $f\left(s\right):={\left(f_{1}\left(s\right),\ldots ,f_{n}\left(s\right)\right)}^{⊤}$.

Since the growth of a single strain in batch experiments is driven by the substrate concentration, when no interactions are present one should recover expression $f_{i}\left(s\right)$. Therefore if there are no interactions then $I_{i}\left(x\right)=1$. From this hypothesis, note that $\lim_{x\to 0}I_{i}\left(x\right)=1$ since if there is minimal presence of OTU, interactions cannot exist. Furthermore for this study it is assumed that $I_{i}\left(\cdot \right)$ is a continuously differentiable function on *x*. For making explicit all of the former:

**Hypothesis** **6** **(H6).**
*The interaction function I previously defined satisfies:*

$I\left({\left(0,\ldots ,0\right)}^{⊤}\right)={\left(1,\ldots ,1\right)}^{⊤}$

*There is an open set $\Omega \subset R^{n}$ such that $I\in C^{1}\left(\Omega \right)$.*



Note $J_{I}\left(x\right)$ the Jacobian matrix of function I, then a first order approximation of I(·) centred at $\bar{x}$ gives: $I\left(x\right)=I\left(\bar{x}\right)+J_{I}\left(\bar{x}\right)\left(x-\bar{x}\right)+o(∥x-\bar{x}∥)$. When $\bar{x}=0$ one recovers the growth expression from the previous section (Equation (6)) implying that $J_{I}\left(\bar{0}\right)$ can be seen as the interaction matrix from model (3).

### 4.1. Unravelling the Interaction Function

Suppose that the functions $f_{i}\left(s\right)$, and the yields $y_{i}$ are well-known. By using experimental measurements of *x*, represented by z(t), the objective is to reconstruct function I(x). For doing so, the terms $I_{i}\left(x\right)$ are replaced by controls $u_{i}\left(t\right)$, thus $I_{i}\left(x\left(t\right)\right)=u_{i}\left(t\right)$. A control law is obtained by solving a nonlinear optimal tracking problem.

Consider the observable system (25), with y(t)=x(t) being the output, because we are observing measurements coming from genetic sequencing.
(25)$$\begin{array}{c} \begin{array}{cc} \dot{x_{i}}= & \left(f_{i}\left(s\right)u_{i}\left(t\right)-D\right)x_{i}\;\forall i\in G_{1}\\ \dot{x_{i}}= & \left(f_{i}\left(s\right)u_{i}\left(t\right)-D\right)x_{i}\;\forall i\in G_{2}\\ \dot{s_{1}}= & \left(s_{in}-s_{1}\right)D+\sum_{i\in G_{1}}y_{s_{1}/x_{i}}f_{i}\left(s\right)u_{i}\left(t\right)x_{i}\\ \dot{s_{2}}= & -s_{2}D+\sum_{i\in G_{1}\cup G_{2}}y_{s_{2}/x_{i}}f_{i}\left(s\right)u_{i}\left(t\right)x_{i}\\ \dot{s_{3}}= & -s_{3}D+\sum_{i\in G_{2}}y_{s_{3}/x_{i}}f_{i}\left(s\right)u_{i}\left(t\right)x_{i}\\ y= & x \end{array} \end{array}$$

Consider the weighted norms defined by positive definite matrices *Q* and *R*, represented by ${∥\cdot ∥}_{Q}$ and ${∥\cdot ∥}_{R}$, respectively, and $\bar{u}>0$. The optimal tracking problem is defined as:(26)$$\begin{array}{c} \begin{array}{cc} \min & \int_{0}^{T}{∥y-z∥}_{Q}+∥\left(u-\vec{1}\right)∥R^{dt}\\ s.t. & (x,s_{1},s_{2},s_{3})\mathrm{solution}\;\mathrm{of}\;(25)\\ \; & u_{i}\left(t\right)\in \left[0,\bar{u}\right] \end{array} \end{array}$$

The control u(t) is intended to drive the system to be near a desired output z(t), which in this context are the measurements of the concentrations of OTU. The term $∥\left(u-\vec{1}\right){∥}_{R}$, was added for two reasons:First, because the interest is testing the idea that interactions could be driving the system. Therefore adding a penalization in the objective function for each control to remain near 1 can be seen as an attempt to explain data without any interaction. In other words, if the control terms are found to drift from 1, it means that interactions are necessary to explain the system dynamics.Second, to force a regularized control. Otherwise note that *u* is linear in (25), therefore if the integral cost does not have a non-linear expression of *u* the optimal control will be of a bang-bang type with possibly singular arcs [25]. Since the objective is to find a differentiable expression of I(x) the addition of the regularization term is deemed necessary.

The problem of approximating the solution of the system to a desired reference (*z* in this case) is called the optimal tracking problem. For solving such a problem the approach developed by Cimen et al. [17,26] was adapted to our problem. The method proposed involves the resolution of Approximating Sequences of Ricatti Equations (ASRE). It consists of iteratively calculating trajectories of System (25) with a certain control law to later feed a non-autonomous Ricatti differential equation with the resulting trajectory. Then, a new control law that uses the solution of the Ricatti equation is proposed and a new trajectory is calculated. The iteration is stopped when a convergence in the output or the control is observed.

The control term should remain positive for the system to be well posed (no negative states), and an upper bound was added to represent the fact that life cannot grow infinitely fast. The tracking problem does not consider a constrained control. Nevertheless, the methods of Cimen et al. [26] were directly used with an explicit constraint in the synthesis of the control. Even though this is probably suboptimal when the control reaches its bounds, one at least is certain about its optimality when the control never reaches its constraints.

Another departure from their method is that in their formulation an approximation of the dynamics for calculating the trajectories is used. They proved that such a linearisation converges to the original dynamics. In the case here presented the linearisation of the dynamics was not necessary.

The change of variable $u_{i}\left(t\right)=v_{i}\left(t\right)+1$ is used for technical reasons explained in Appendix C.

This in turn implies $v_{i}\left(t\right)\in \left[-1,\bar{u}-1\right]$. The feedback control v(t), will be of the form $v\left(t\right)=-R^{-1}{\tilde{B}}^{⊤}\left(t\right)\left(\tilde{P}\left(t\right)x\left(t\right)-s_{f}\left(t\right)\right)$. Where matrix P(t) and vector $s_{f}\left(t\right)$ solve differential equations. In Appendix C it is proved that, thanks to the structure of the system, one only needs to calculate 2n differential equations for the synthesis of the control, instead of ${\left(n+3\right)}^{2}+\left(n+3\right)$ that would imply the direct application of the method, which renders the method—at least theoretically—scalable for a growing number of OTU.

### 4.2. Proof of Concept

The approach was tested with data generated by simulating model (3) using the parameters of the case study 1 (Table 2) with interaction matrix given by (Figure 1b). In the operating diagram of the same case (Figure 2b) the red zone implies complete nitrification, while the green zone means partial nitrification. For integrating the former phenomena in the simulation, the system was simulated for 300 days, and perturbed at day 150 from the CN zone ($\left(s_{in},D\right)=\left(1.25,0.24\right)$) to a PN zone ($\left(s_{in},D\right)=\left(1.95,0.24\right)$). Simulations can be seen in Figure 5. Note how from day 150 the NOB population (OTU 2) represented in Figure 5b decreases, which in turn implies a decrease in $s_{3}$, as seen in Figure 5c. In the case where no interactions take place, the OD seen in Figure 2a implies that $s_{3}$ would have accumulated all along the trajectory, since the perturbation still remains in the CN zone.

For the tracking procedure the functions $f_{i}\left(s\right)$ and the yields $y_{i}$ were the same as those used for simulating the synthetic data (parameters in Table 2) and the control is meant to account for the interaction term. The *Q* and *R* matrices were $\left[\begin{array}{cc} \lambda_{1}I_{n_{1}} & 0\\ 0 & \lambda_{2}I_{n_{2}} \end{array}\right]$ and $I_{n}$, respectively, with $\lambda_{1}=10^{-4}$ and $\lambda_{2}=10^{-5}$ in order to better track the NOB trajectories, since they are less abundant. The values were obtained by trial and error, by using a single λ for both functional groups, beginning with λ=1, in which case one can see how the optimal control becomes u=1, thus no tracking is performed. Further diminishing the value from $10^{-5}$ adds too much noise to the control, without significant gains on the quality of the tracking.

The results of the procedure to identify interactions can be seen in Figure 6 and Figure 7. Figure 6a and Figure 7a,b show the total biomass concentration, and the trajectories for the OTU belonging to $G_{1}$ and $G_{2}$, respectively. It can be seen that the method approaches well the trajectories of the OTU, with a better result for the AOB community, which can be explained by the one order of magnitude difference in their concentrations (which in turn is a consequence of the one order of magnitude difference in their yields). The metabolites concentration represented in Figure 6b are in accordance with the simulated: The method is able to reconstruct the metabolites trajectories from the community measurements.

Figure 8 and Figure 9 show the controls and the corrected growth rate for each functional group. The control for each functional group can be seen in Figure 8a and Figure 9a. Note that from the structure of a quadratic regulator, since there is no cost in the final state, the end value is always 1. Figure 8b and Figure 9b show the resulting growth rates for AOB and NOB, respectively, without the control $u_{i}\left(t\right)$. Figure 8c and Figure 9c are the complete expression that determines growth rate, that is $f_{i}\left(s\left(t\right)\right)u_{i}\left(t\right)x_{i}\left(t\right)$. Note how little the shape changes with respect to Figure 8b and Figure 9b, which might mislead the reader to conclude that the control had reduced effects in the dynamics. The way out of this conundrum is to remember that the control’s effects are already included in $x_{i}\left(t\right)$ and $s_{i}\left(t\right)$, and thus in expression $f_{i}\left(s\left(t\right)\right)x_{i}\left(t\right)$.

A final comment on the identifiability of the interaction terms. Even though one might propose a growth rate with the tracking control u(t) that accurately replicates the OTU trajectory x(t), retrieving the original interaction coefficients from the obtained control for this example was not possible. The former was tried by minimizing function $f\left(A\right)=∥\int_{0}^{T}u\left(t\right)-\left(1+Ax\left(t\right)\right)dt∥$ with a non linear optimization solver for a 1000 initial random guesses for matrix *A*. If one also takes into account ${\bar{\mu }}_{i}$ and $K_{i}$ as parameters to fit this adds even more degrees of freedom, thus suggesting that the identifiability of growth functions (6) in model (3) might be very low.

## 5. Application

The tracking problem was applied to data coming from a nitrification process with experimental conditions described in [27]. For exploring the hypothesis of interactions as drivers of bioreactors performance environmental conditions should be kept as constant as possible. Therefore only data from day 183 onwards was used because a change in the operating temperature happened at that point, which is known to have an effect on kinetics. For choosing which species belong in which functional group, the procedure described of Ugalde-Salas et al. [28] was used. From day 183 to day 315, 31 OTU were identified in the $G_{1}$ group (AOB) and 5 in the $G_{2}$ functional group (NOB).

A first example of the procedure is performed when the classified OTU are regrouped in their assigned functional groups by adding their concentrations. A 5 dimensional dynamical system is obtained, thus there are only two interacting functional biomasses: this case is structurally the same as in the proof of concept, but here a real dataset is used. The same procedure is applied where no regrouping occurs and the system state grows to 39.

The knowledge of functions $f_{i}\left(s\right)$ was based on a study of nitrification’s kinetic parameters [29]. Particularly given the system’s ammonium and nitrite concentration a Monod function (Equation (27)) was used for $G_{1}$ and $G_{2}$ with parameters given in Table 4 calculated from the equation of Table 2 of the same article. The yields were fitted to match the nitrogen mass balances. The *Q* and *R* matrices were the same as in the proof of concept section, that is $\left[\begin{array}{cc} \lambda_{1}I_{n_{1}} & 0\\ 0 & \lambda_{2}I_{n_{2}} \end{array}\right]$ and $I_{n}$, respectively, with $\lambda_{1}=10^{-4}$ and $\lambda_{2}=10^{-5}$, because data lie in the same order of magnitude than synthetic data.
(27)$$\begin{array}{cc} f_{i}\left(s\right) & ={\bar{\mu }}_{2}\frac{s_{1}}{K_{1}+s_{1}}\;\forall i\in G_{1}\\ f_{i}\left(s\right) & ={\bar{\mu }}_{2}\frac{s_{2}}{K_{2}+s_{2}}\;\forall i\in G_{2} \end{array}$$

For the reader to gain understanding of the situation, a simulation of the system using the experiments operating parameters (*D* and $s_{in}$) is presented without control (i.e., u(t)=1) in Figure 10 nitrate ($s_{3}$) accumulates all along the trajectory, but when compared to data it is clear that $s_{3}$ stops accumulating after a while.

When applying the tracking method one obtains the simulation that can be seen in Figure 11 and Figure 12. The method captures the tendencies of the measured substrates as seen in Figure 11b. The tracking of each functional group $G_{1}$ (AOB), and $G_{2}$ (NOB) can be seen in Figure 12a,b, respectively.

The growth rates of each functional group are shown in Figure 13. Note in the case of AOB (Figure 13a) the resulting growth rate shows a noisy curve formed by pulses. The behaviour of the NOB community (Figure 13b) is qualitatively very similar with somewhat stronger pulses and less noise. The former is to be expected since more OTU were regrouped to compose the AOB biomass, therefore more noise sources were added.

The same procedure is applied without regrouping. The results on total biomass and metabolites are shown in Figure 14a,b, respectively. Both patterns still fit the data, but to a lesser degree of precision when compared to Figure 11. This can be explained by inspecting Figure 15, Figure 16, Figure 17, Figure 18, Figure 19 and Figure 20. First note the absolute error of the tracking for each of the OTU in the AOB community (Figure 15b, Figure 16b, Figure 17b, Figure 18b and Figure 19b), almost every point lies below 0.015 [g/L], implying that the method might not be able to track below that threshold for the members of the AOB community. The former ultimately implies that the most abundant OTU are better tracked, thus the information contained in the least abundant species is not integrated in the model. Notice that the error for the NOB community is lower (Figure 20b), almost every point lies below 0.005 [g/L], this can be explained in the one order of difference in the entry of matrix *Q* for the AOB and the NOB community. It may be the case that using appropriate weight matrices that account for the difference between OTU abundances could help in this aspect; in that sense only one rational was tested (inverse of the mean abundance of each OTU in the diagonal entries of matrix *Q*) and did not improve the results. When looking at the growth rates (Figure 15c, Figure 16c, Figure 17c, Figure 18c, Figure 19c and Figure 20c) one again observes pulses for each OTU. Finally, note that most OTU were present only for a fraction of the experiment’s duration.

In both cases, the regrouped and individual tracking, the growth rate varies strongly, raising the question whether the observed pulses are emerging from interactions within the microbial community. When growth rates are compared to the proof of concept section it seems doubtful that a linear pairwise interaction model such as the gLV model could capture the complexity of the particular chemostat analysed. Perhaps these interactions are not constant through time (as opposed to the gLV model) or a different interaction function should be thought of. However the former questions cannot be fully clarified here, because the quality of the genetic sequencing from molecular fingerprints might not be the best when compared to more recent techniques, thus it is unclear if the pulses are due to noise of the measurements.

The interpretation of the correction term as interactions is not the only possible reading. In other contexts the correction term might also be interpreted as a non accounted phenomena ranging from environmental factors (e.g., temperature, pH) to other biological factors (viruses, flock formation, pathogens). Alternative hypotheses for explaining the observed patterns in the microbial community should be considered as well.

## 6. Conclusions and Perspectives

Over the last decades, advances in genetic sequencing and microbial ecology have opened a gap for modellers in biochemical processes to integrate this valuable information. Considering the success of mass balance models to predict and pilot bioreactors, new models should be built upon them, or at least be compared against them. This article exposed what can be gained from combining population-based models as used in ecology with functional group based approaches as used in bioengineering. The analysis of the gLV model proposed by Dumont et al. [16] already shows that such a combination can give way to models that include bi-stability, coexistence within a functional group, and unintuitive operational insights such as raising the input ammonium $s_{in}$ to achieve partial nitrification. The increased number of parameters of this particular model obviously hinders its potential application, but it surely helps to illustrate what can be gained by joining both types of models. The mathematical analysis focused on the particular case of pairwise interactions, which can be seen as a first order approximation of the introduced concept of interaction function. This opens the question for a broader class of interaction functions that could well represent complex microbial ecosystems, particularly bioreactors.

With that line of reasoning, in order to understand what this interaction function should look like, a data-driven approach was presented. It can be simply described as correcting the growth rate expression of each individual species in a mass balance model by explicitly assuming a control loop on the growth rate depending on the species state variables and the measured abundances. The reconstructed growth rates seem to consist of pulses, suggesting that a form of a possible interaction function should reproduce this behaviour. However, the former questions cannot be fully clarified here, because the identification and quantification of OTU from single-strand conformation polymorphism (SSCP) as performed by Dumont et al. [18] might not be optimal. Indeed, the determination of OTU from SSCP profiles rely on the identification and quantification of peaks and miss many OTU that appear only as background noise [30]. This type of fingerprinting is thus less accurate than more recent methods such as the sequencing of the 16S ribosomal RNA. In spite of this, one is able to recover the substrates dynamics, implying that the hypothesis of microbial interactions as drivers of a bioprocess, in the form of feedback loops affecting each others growth rates, is not far-fetched.

The use of the tracking technique can be applied in a straightforward manner to already existing models in mixed homogeneous bioreactors, under the condition that the microbial species have already been identified with a particular functionality of the system. Even though the tracking model was proposed for a chemostat setting as a way to correct a substrate limited expression, the method could be used in contexts where less information on the growth function of microbes is known. If one supposes nothing on the growth expression but the fact that is bounded (life cannot grow infinitely fast) the model becomes a linear model and one recovers a classic quadratic regulator for linear systems. This was tested for the data presented in this article and, unfortunately, negative substrate appeared as an output, suggesting that the substrate limitation term is crucial for the model to be well-posed. Nevertheless, even in the former case, the synthesis of the optimal bounded control remains an open theoretical challenge. One might bypass this issue of the current control scheme by, for example, a very thorough use of the Pontryagin maximum principle for the synthesis of the control. In a more general view the reconstruction of the growth function in chemostat systems is already subject to problems of identifiability [31], integrating genetic sequencing could provide a path for more certainty in model calibration.

## Figures and Tables

**Figure 1 bioengineering-08-00031-f001:**
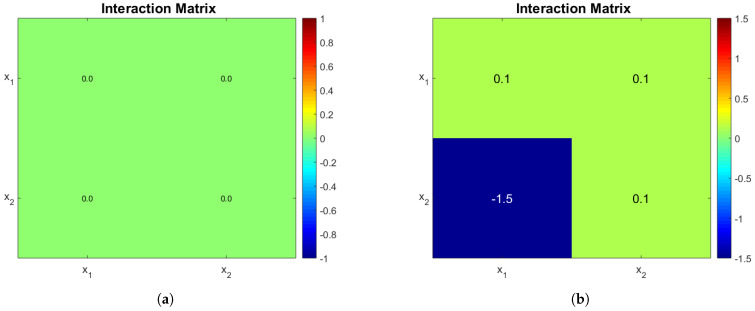
Interaction matrices. Note how the presence of *x*_1_ affects very negatively *x*_2_ in (**b**), with respect to other interactions. The terms in the diagonal entries of the matrix represent intraspecies interactions, while the terms off the diagonal represent the interspecies interactions. (**a**) Interaction matrix of model (3) with no interactions. (**b**) A non-zero interaction matrix of model (3).

**Figure 2 bioengineering-08-00031-f002:**
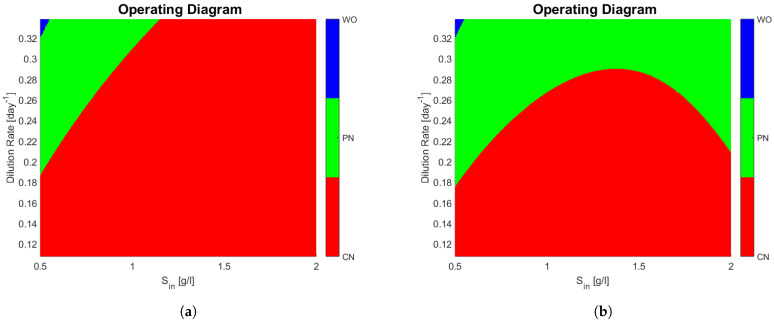
(**a**) Operating diagram of model (3) with no interactions (interaction matrix represented by Figure 1a). (**b**) Operating diagram of model (3) with interactions represented by Figure 1b. Note how (**b**) has a much larger zone where partial nitrification takes place. This is due to the negative interaction of *x*_1_ on *x*_2_.

**Figure 3 bioengineering-08-00031-f003:**
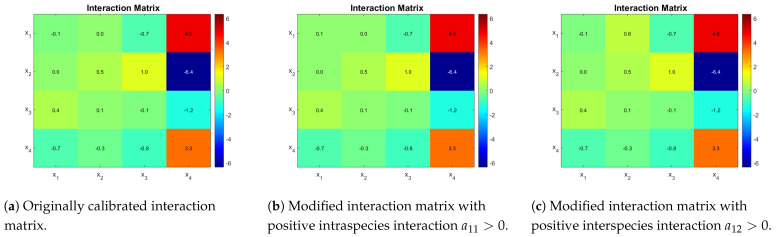
Interaction matrices for each case for a consortia of 4 bacterial species where *x*_1_ and *x*_2_ are AOB and *x*_3_ and *x*_4_ are NOB. Parameters *a*_11_ and *a*_12_ were modified in (**b**,**c**), respectively.

**Figure 4 bioengineering-08-00031-f004:**

Ecological diagrams (ED). The different zones represent the combination of surviving species in the steady state. PN takes place when neither *x*_3_ nor *x*_4_ are present. CN takes place if *x*_3_ or *x*_4_ are present. (**a**) ED from interaction matrix on Figure 3a. (**b**) ED from interaction matrix on Figure 3b. In the legend (1) and (2) represent the two different stable equilibria in each zone. (**c**) ED from interaction matrix on Figure 3c. Note that in (**b**) two stable equilibria exist for each zone.

**Figure 5 bioengineering-08-00031-f005:**
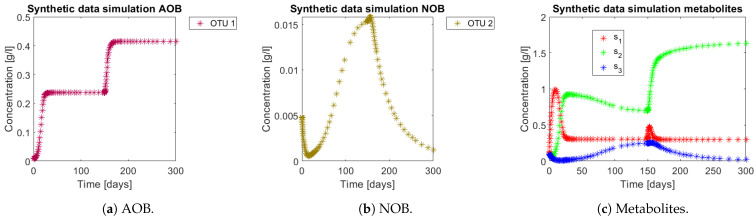
Synthetic data generated by model (3), with parameters from case study 1. Note the effects of the increased input *_s_in__* generated in day 150.

**Figure 6 bioengineering-08-00031-f006:**
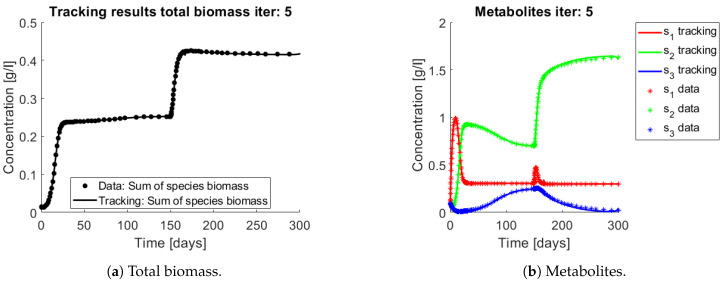
Asterisks represent the synthetic data, while the continuous lines represent the method’s output. The method is able to reconstruct the metabolites pattern, from the biomasses concentrations.

**Figure 7 bioengineering-08-00031-f007:**
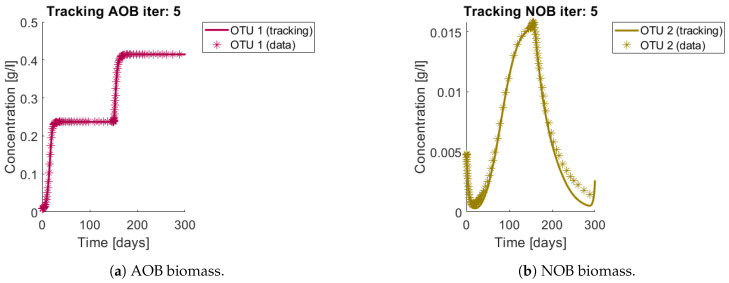
Asterisks represent the synthetic data, while the continuous lines represent the method’s output. The method reconstructs a continuous trajectory from the synthetic data.

**Figure 8 bioengineering-08-00031-f008:**
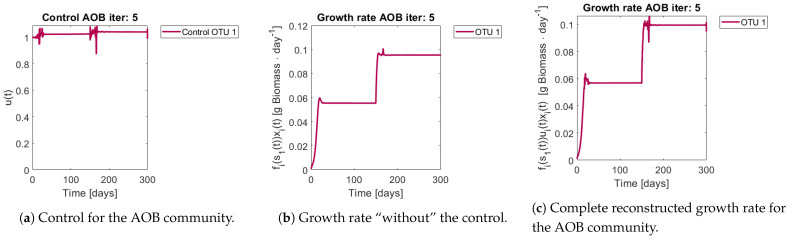
Obtained control and reconstructed growth rate for OTU 1 (AOB).

**Figure 9 bioengineering-08-00031-f009:**
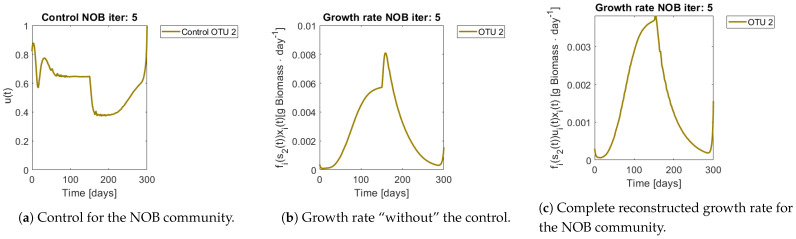
Control and reconstructed growth rate for OTU 2 (NOB).

**Figure 10 bioengineering-08-00031-f010:**
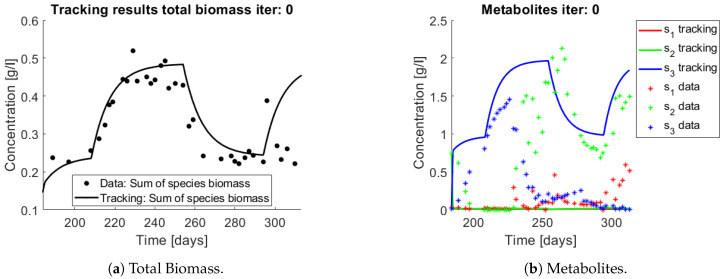
Simulation of system (25) when *u* = 1, with functions as in (27). Data points are represented by a star. The continuous line represents the simulation.

**Figure 11 bioengineering-08-00031-f011:**
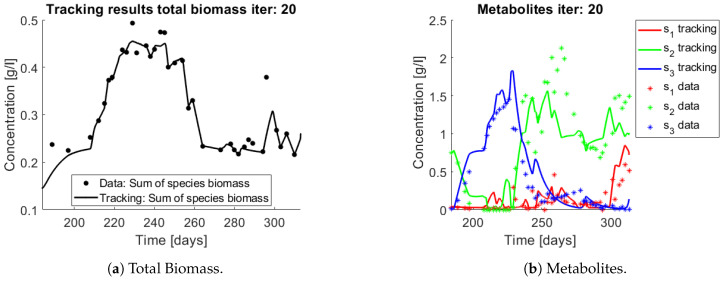
Results on applying the tracking method to a nitrification experiment when regrouping OTU in their functional groups. Data points are represented by asterisks. The continuous line represents the tracking procedure results.

**Figure 12 bioengineering-08-00031-f012:**
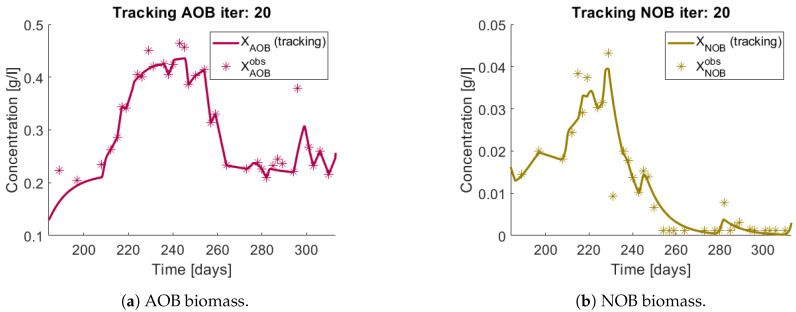
The tracking procedure applied to the observed biomass (asterisks) regrouped in two functional groups.

**Figure 13 bioengineering-08-00031-f013:**
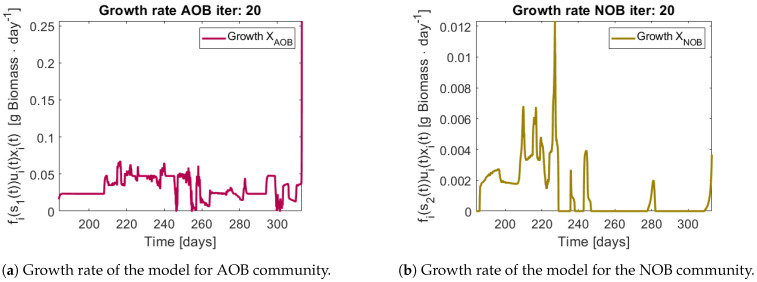
Obtained growth rates when regrouping OTU in their functional groups.

**Figure 14 bioengineering-08-00031-f014:**
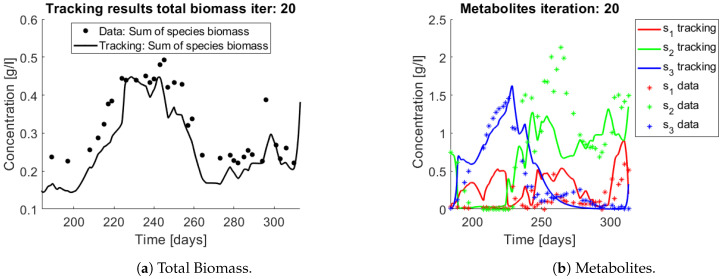
Results on applying the tracking method to a nitrification experiment when all OTU are tracked independently. Data points are represented by a star. The continuous line represents the tracking procedure results.

**Figure 15 bioengineering-08-00031-f015:**
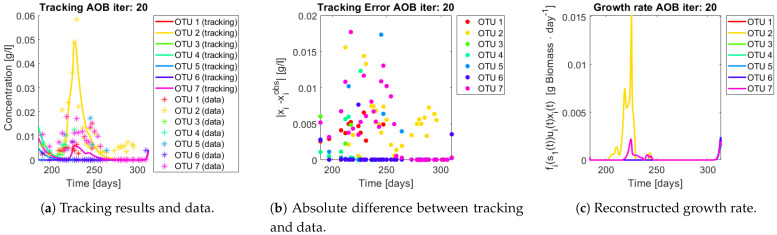
Results for OTU 1-7 (AOB).

**Figure 16 bioengineering-08-00031-f016:**
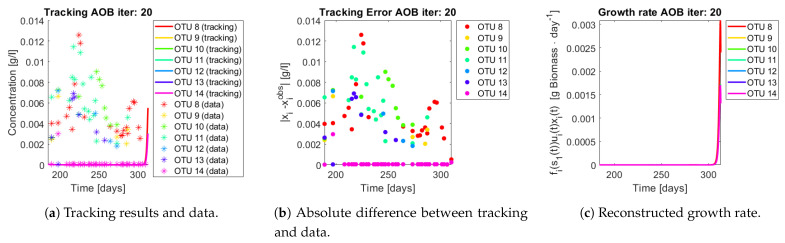
Results for OTU 8-14 (AOB).

**Figure 17 bioengineering-08-00031-f017:**
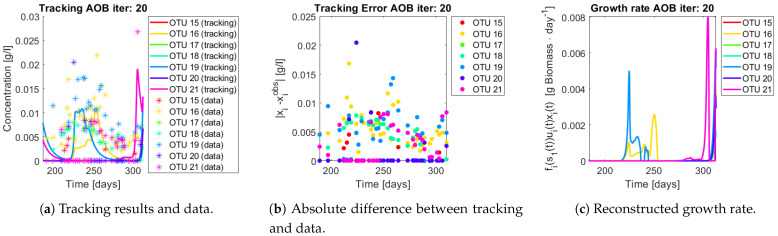
Results for OTU 15-22 (AOB).

**Figure 18 bioengineering-08-00031-f018:**
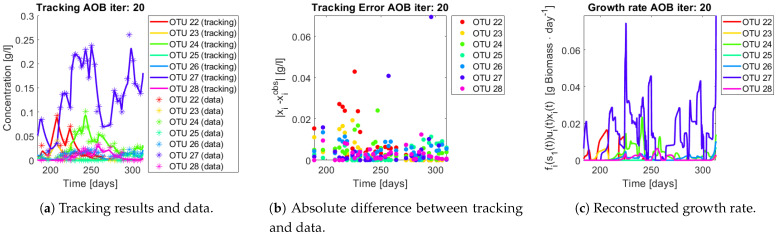
Results for OTU 23-28 (AOB).

**Figure 19 bioengineering-08-00031-f019:**
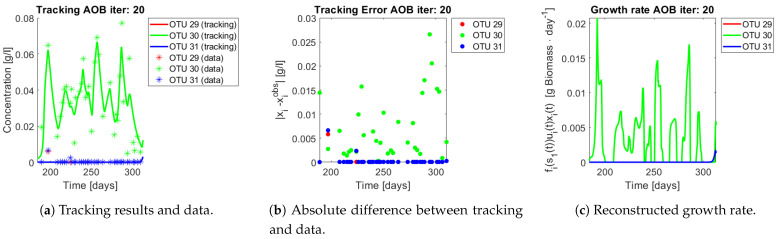
Results for OTU 29-31 (AOB).

**Figure 20 bioengineering-08-00031-f020:**
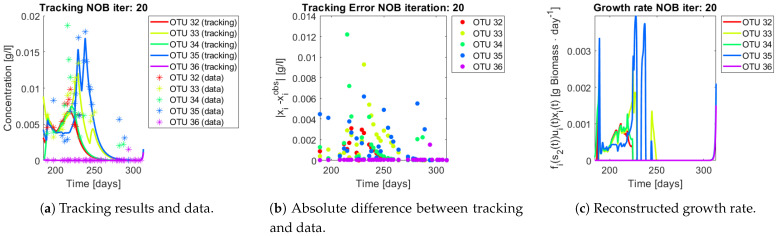
Results for OTU 32-36 (NOB).

**Table 1 bioengineering-08-00031-t001:** Relationship of $y_{s_{i}/x_{j}}$ with $y_{j}$.

Yields per Biomass Formed	$j\in G_{1}$	$j\in G_{2}$
$y_{s_{1}/x_{j}}$	$-\frac{1}{y_{j}}$	0
$y_{s_{2}/x_{j}}$	$\frac{1}{y_{j}}$	$-\frac{1}{y_{j}}$
$y_{s_{3}/x_{j}}$	0	$\frac{1}{y_{j}}$

**Table 2 bioengineering-08-00031-t002:** A set of kinetic parameters of model (3).

Kinetic Parameters	${\bar{\mu }}_{i}$ [1/day]	$K_{i}$ [g/L]	$\frac{1}{y_{i}}$ [gr/gr]
$x_{1}\in G_{1}$	0.77	0.7	3.98
$x_{2}\in G_{2}$	1.07	0.3	16.12

**Table 3 bioengineering-08-00031-t003:** Kinetic parameters of model (3) from Dumont et al. [16].

Case Study 2 Kinetic Parameters	$\mu_{i}$ [1/day]	$K_{i}$ [mg/L]	$\frac{1}{y_{i}}$ [gr/gr]
$x_{1}\in G_{1}$	0.828	0.147	3.85
$x_{2}\in G_{1}$	0.828	0.147	3.85
$x_{3}\in G_{2}$	0.18	0.026	100
$x_{4}\in G_{2}$	0.18	0.026	100

**Table 4 bioengineering-08-00031-t004:** A set of kinetic parameters of model (25).

Kinetic Parameters	$\mu_{i}$ [1/day]	$K_{i}$ [g/L]	$\frac{1}{y_{i}}$ [gr/gr]
$x_{1}\in G_{1}$	1.97	$7\times 10^{-1}$	4.49
$x_{2}\in G_{2}$	1.87	$5.4\times 10^{-1}$	45.51

## Data Availability

Data can be recovered from the following repository https://github.com/paus-5/Class-and-Track (accessed on 24 February 2021).

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
