# Peer review of "Microbial Interactions as Drivers of a Nitrification Process in a Chemostat"

_bioengineering, 2021, doi:10.3390/bioengineering8030031_

Round 1

Reviewer 1 Report

The manuscript by Ugalde-Salas et al. modelled bioprocesses in a chemostat, with the focus on microbial interaction. While the model construction and algorithm development are beyond my expertise, I find this work valuable in terms of its application in designing and monitoring bioreactors. Below are a few comments that could hopefully increase the accessibility of this manuscript to a broader spectrum of readers in the field of bioengineering and microbial ecology. 

Line 104. The notation of xi (with a dot above) should be explained. Does it mean the rate of change of xi? I am also confused by the expression of s1 (with a dot above it). If it means changing rate of s1, the first term on the right-hand side represents how much s1 physically flows out of the reactor, and the second term represents how much s1 is biologically consumed by x. Shouldn’t the terms added up to represent the total change rate?

Line 108. Pairwise interaction seems the key component of this study. It would be helpful to explain here, where it is defined for the first time, why it can be expressed so. Why are aijxj summable? Why it adds with 1? What is the value range of this term and what is the biological inference? Such an explanation of the interaction matrix can greatly facilitate reading the results. Maybe the authors can also consider elaborating the ecological and mathematical concept of interaction in Introduction to better prepare readers for the results. On top of that, the Introduction could include more elaboration on the novelty of this research and the advantage of the presented method.

Line 217. It is quite clear from the result that interaction led to different nitrification performance. What is the biological implication of interaction? That could better help readers to understand the results in figure 1.

Figure 12. This figure tries to display the tracking results of individual OTUs. I am wondering whether it will be more informative to plot the differences between modelled and measured OTU concentration? 

Reviewer 2 Report

The manuscript presents a somewhat novel approach for modelling bioprocesses using a standard kinetic framework based on a generalised Lotka-Volterra model with an interaction matrix to allow for discrimination amongst multiple (n) OTUs, potentially identified via molecular methods (i.e. sequencing). This part is based on previous work conducted by Dumont et al. However, the main novelty of the presented manuscript is the use of control theory; namely, constrained feedforward-feedback control that is able to track and correct the terms in the kinetic growth model (resulting in potential periodicity/pulses). Unfortunately, the latter was not completely successful but I found the approach to be interesting and worthy of documentation and further study.

Overall I found the manuscript to be well written but very top-heavy on mathematics. While I understand the necessity for documenting the analysis in rigorous way, I am unsure whether the readers of bioengineering would have the necessary patience to plough through the fairly dense nomenclature (from a non-expert viewpoint, I recognise the analyses performed here is not complex for a biomathematician). Nevertheless, I cannot make any recommendation for moving the definition-lemma-proof sections to supplementary material and am happy that it adds weight as part of the main text. 

Overall a good paper that is frustrating in that the work has not reached a satisfactory conclusion. However, the approach looks promising for further research building on this manuscript.

Some minor notes.

For note, AOB – Ammonia oxidising bacteria, not ammonium (throughout)

Line 2 (and elsewhere): Operative -> Operational

Line 25: ‘it is hard to define precisely the …’

Line 28: provide the acronym (FISH)

Line 133: A short note on why analytical derivation of stability is not performed should be stated here

Figures 6 & 7: the captions are partly obscured by the images

Line 411: Change to “Over the last decades, advances…”

Line 412-413: Why? (why should new models be built upon them?)

Line 421-422: change to “of an interaction function.”

Line 422-423: change to “well represent complex…”

Line 424: change to “With that line of reasoning…”

Line 426: remove the comma after model

Line 429: what is meant by the quality of molecular fingerprints? Do you mean the resolution of molecular techniques for characterising the dynamics of microbial systems in complex environments is difficult and not yet resolved to allow for this insight?
